# CALMODULIN1 and WRKY53 Function in Plant Defense by Negatively Regulating the Jasmonic Acid Biosynthesis Pathway in Arabidopsis

**DOI:** 10.3390/ijms23147718

**Published:** 2022-07-13

**Authors:** Chunyang Jiao, Kaixiang Li, Yixin Zuo, Junqing Gong, Zhujuan Guo, Yingbai Shen

**Affiliations:** 1College of Biological Sciences and Technology, Beijing Forestry University, Beijing 100083, China; chunyangjiao@126.com (C.J.); lkx202@126.com (K.L.); zuoyx2020@163.com (Y.Z.); 13120030501@sohu.com (J.G.); gzj2190073@163.com (Z.G.); 2National Engineering Research Center of Tree breeding and Ecological Restoration, College of Biological Sciences and Technology, Beijing Forestry University, Beijing 100083, China; 3Guangxi Key Laboratory for Cultivation and Utilization of Special Non-Woods Forest, Guangxi Forestry Research Institute, Nanning 530002, China

**Keywords:** Arabidopsis, *Spodoptera littoralis*, CAM1-WRKY53, jasmonic acid

## Abstract

Jasmonic acid (JA) is an important hormone that functions in plant defense. *cam1* and *wrky53* mutants were more resistant to *Spodoptera littoralis* than in the wild-type (WT) Arabidopsis group. In addition, JA concentration in *cam1* and *wrky53* mutants was higher compared with the WT group. To explore how these two proteins affect the resistance of Arabidopsis plants, we used a yeast two-hybrid assay, firefly luciferase complementation imaging assay and in vitro pull-down assay confirming that calmodulin 1 (CAM1) interacted with WRKY53. However, these two proteins separate when calcium concentration increases in Arabidopsis leaf cells. Then, electrophoretic mobility shift assay and luciferase activation assay were used to verify that WRKY53 could bind to lipoxygenases 3 (LOX3) and lipoxygenases 4 (LOX4) gene promoters and negatively regulate gene expression. This study reveals that CAM1 and WRKY53 negatively regulate plant resistance to herbivory by regulating the JA biosynthesis pathway via the dissociation of CAM1-WRKY53, then the released WRKY53 binds to the *LOXs* promoters to negatively regulate *LOXs* gene expression. This study reveals WRKY53′s mechanism in insect resistance, a new light on the function of WRKY53.

## 1. Introduction

Plant resistance to insect feeding can be improved through the following aspects: perception of herbivorous insects; excitation of early signal; hormone-mediated signal transduction; metabolism (mainly the synthesis of anti-insect substances); and phenotypic changes [1].

Arabidopsis shows an increase in cytosolic calcium concentration following *Spodoptera littoralis* herbivory in leaves [2]. However, plant signaling pathways are complex and different members in the same family may cause opposite effects. For example, CML37 and CML42 are two Ca^2+^ ion response factors that regulate JA biosynthesis to change the tolerance to insects. CML42 is a positive regulator, while CML37 is a negative regulator [3,4]. Calmodulins (CAMs) also act as important calcium sensors and play important roles in plant stress, such as AtCaM3 which regulates heat shock, AtCaM4 participates in salt stress and negatively affects freezing tolerance [5,6,7,8,9]. However, it is not clear whether CAMs are involved in JA-regulated Arabidopsis plant defense against *Spodoptera littoralis*.

JA is a hormone found widely in plants that function in defense responses [10]. 13-Lipoxygenases are the initial enzymes in the JA biosynthesis pathway [11]. Arabidopsis contains four 13-lipoxygenase genes. The 13-lipoxygenases LOX2, LOX3 and LOX4 catalyze the conversion of α-linolenic acid into 13(*S*)-hydroperoxide [12,13]. Among these, *LOX3* and *LOX4* have W-boxes (the DNA sequence (C/T)TGAC(T/C)) in their promoters and interact with WRKYs.

Many WRKYs participate in the JA biosynthesis pathway. NaWRKY3 and NaWRKY6 improve JA and JA-Ile levels by increasing the expression of JA biosynthesis genes, including *LOX*, *AOS*, *AOC*, and *OPR* [14]. *HvLOX1* in barley (*Hordeum vulgare*) contains W-boxes in its promoter and appears to interact with WRKY transcription factors [15]. In Arabidopsis, the binding of JAV1 and WRKY51 to the *AOS* promoter depends on the presence of a W-box [16]. The Arabidopsis genome encodes 74 WRKYs, which are divided into three groups based on the number of WRKY domains and the pattern of the zinc-finger motif [17,18]. WRKY53 participates in many signaling pathways in Arabidopsis, including leaf senescence, drought stress, abiotic stress and biotic stress signaling [19,20,21,22,23]. However, whether WRKY53 promotes the JA biosynthesis pathway remains unexplored.

Thus, in the current study, we demonstrated how CAM1 and WRKY53 negatively regulate Arabidopsis resistance according to a yeast two-hybrid assay, firefly luciferase complementation imaging assay, in vitro pull-down assay, electrophoretic mobility shift assay and luciferase activity assay. CAM1-WRKY53 complex breaks down due to high calcium concentration, then WRKY53, which detaches from the CAM1-WRKY53 complex, negatively regulates JA content by negatively regulating *LOXs* gene expression.

## 2. Results

### 2.1. CAM1 and WRKY53 Maybe Negatively Regulate the JA Biosynthesis Process When Resistant to Spodoptera Littoralis

Because *Spodoptera littoralis* herbivory increased calcium concentrations in Arabidopsis mesophyll cells, we reasoned that CAMs might take part in Ca^2+^ signal transduction. Many studies have shown that CAM1 functions in plant resistance. CAM1 interacts with and activates GAD1, leading to the accumulation of GABA, which is involved in many plant stress responses [24,25]. CAM1 is also involved in abscisic acid-mediated ROS accumulation, stomatal closure, and leaf senescence [26]. Furthermore, changes in *CAM1* and *CAM4* expression lead to changes in NO levels in Arabidopsis leaves [8]. By contrast, CAM2 is involved in pollen development [27], CAM3 mainly participates in the heat shock response in Arabidopsis and CAM7 promotes photomorphogenesis [6,28]. Therefore, we selected CAM1 for further analysis.

Since CAMs interact with group II WRKYs and group III WRKY53 [29,30] and WRKYs activate *LOXs* transcription, we measured five group II WRKY gene (*WRKY11*, *WRKY15*, *WRKY17*, *WRKY21*, *WRKY39*) expression, and *WRKY53* gene expression in WT Arabidopsis and the *cam1* mutant (Appendix A). Only *WRKY53* was significantly expressed in the *cam1* mutant compared with that in WT Arabidopsis.

To confirm the effect of CAM1 and WRKY53 in Arabidopsis on the resistance of *Spodoptera littoralis*, we inoculated WT, *cam1* and *wrky53* Arabidopsis plants with *Spodoptera littoralis* larvae and the WT group was the control group. After 7 days, we measured the length and weight of *Spodoptera littoralis* larvae. Both growth indicators were lower in *cam1* and *wrky53* plants at different levels than in control plants (Figure 1a–c). Additionally, the *cam1* and *wrky53* mutant leaves were also eaten less than WT plants (Figure 1d). Because JA plays an important role in plant defense, we tested JA concentration in WT, *cam1* and *wrky53* plants. The results showed that JA concentration was significantly higher in *cam1* plants and *wrky53* plants than in WT plants. (Figure 1e). These results indicate that JA is important in plant defense against *Spodoptera littoralis* in Arabidopsis, CAM1 and WRKY53 maybe negatively regulate this JA biosynthesis process according to the changes in JA concentration. To further investigate this hypothesis, we conducted the following experiments.

### 2.2. CAM1 Interacts with WRKY53

Because the growth of the insect in both mutants was reduced and the JA content was increased in *cam1* and *wrky53* plants, we reasoned that the roles of CAM1 and WRKY53 are closely related. WRKYs interact with various CAMs [29,30]. To determine whether CAM1 and WRKY53 interact, we carried out yeast two-hybrid (Y2H), firefly luciferase complementation imaging (LCI) and in vitro pull-down assays. Since WRKY53-BK showed self-activation in the Y2H assay, we used the N-terminus (residues 1–217) of WRKY53 as the bait protein and CAM1 as the prey protein. CAM1 and WRKY53 interacted in the Y2H system, as confirmed in three independent experiments (Figure 2a). To further confirm this interaction, we injected *N. benthamiana* leaves with different combinations of Agrobacterium cultures harboring CAM1-Cluc and WRKY53-Nluc in four separate quadrants of each leaf (Cluc/Nluc, CAM1-Cluc/Nluc, Cluc/WRKY53-Nluc and CAM1-Cluc/WRKY53-Nluc). Leaf quadrants harboring CAM1-Cluc/WRKY53-Nluc had stronger fluorescent signals than the other three quadrants (Figure 2c). CAM1 also interacted with WRKY53 in an in vitro pull-down assay. It showed that both CAM1–GST and GST combined with glutathione beads, once HIS fusion protein is bound to these proteins, it can be shown in the anti-HIS pulldown. Additionally, anti–HIS pulldown showed that WRKY53–HIS interacted with CAM1–GST in the pull–down assay, whereas WRKY53–HIS failed to interact with GST (Figure 2b). These results indicate that CAM1 interacts with WRKY53.

For example, AtWRKY7, a typical group IId WRKY protein, contains a conserved CaM-binding domain (CaMBD) with the amino acid sequence 72-VAVNSFKKVISLLGRSR-88 [29]. WRKY53 is a WRKY group III WRKY protein. Compared with the amino acid sequence of WRKY7, WRKY53 also contains a conserved CaMBD with the amino acid sequence 57-VKQIVSSYERSLLLLNW-73: 57 V, 67 S, 68 L and 69 L are conserved (Figure 2d). To determine whether these conserved amino acids play key roles in the interaction between CAM1 and WRKY53, we mutated these amino acids to R. Only the simultaneous mutation of all four amino acids significantly affected the binding of CAM1 and WRKY53 (Figure 2e), which suggested that 57 V, 67 S, 68 L and 69 L form a fixed structure that combines with CAM1.

### 2.3. WRKY53 Negatively Regulates LOX3 and LOX4 Expression

Analysis of hormonal data from the control groups indicated that the levels of JA were higher in *cam1* and *wrky53* than in WT plants (Figure 1e). This result indicates that CAM1 and WRKY53 negatively regulate the JA biosynthesis pathway. Arabidopsis contains four LOXs, which function in the first steps of the JA biosynthesis pathway. The promoters of *LOX3* and *LOX4* each contain W-boxes, a motif recognized by WRKY proteins. Therefore, to further clarify the role of WRKY53 in regulating the JA biosynthesis pathway, we performed EMSA to detect interactions between WRKY53 and the W-boxes in the *LOX3* and *LOX4* promoters and confirmed that WRKY53 bound to the W-boxes in the promoters of both genes (Appendix A). Because the binding between WRKY53 and *LOX3* was stronger than that between WRKY53 and *LOX4*, we focused on *LOX3* in subsequent analysis; the results for *LOX4* are shown in Appendix A.

In a luciferase activity assay, WRKY53 pGreenII 62-SK interacted with the *LOX3* promoter pGreenII 0800-LUC and WRKY53 pGreenII 62-SK negatively regulated *LOX3* pGreenII 0800-LUC expression. However, when CAM1 pGreenII 62-SK was added to the system, *LOX3* pGreenII 0800-LUC expression returned to the level of the control group (Figure 3a). To further explore the roles of calcium and CAM1 in the binding of WRKY53 to the *LOX3* promoter, we conducted two experiments. First, we performed an EMSA using the first W-box in the *LOX3* promoter. When CAM1-MBP was added to the assay system, the interaction between WRKY53-MBP and the W-box decreased. However, when CAM1-MBP and calcium were added to the assay system simultaneously, the results were similar to those of samples containing only WRKY53-MBP and the W-box (Figure 3b). Second, we performed a pull-down assay, which showed that calcium affected the bonding strength of the CAM1- WRKY53 complex (Figure 3c). These results indicate that in healthy leaves (Ca^2+^ in lower concentration), CAM1 is combined with WRKY53, whereas WRKY53 is rarely bound to the *LOX3* promoter. When *Spodoptera littoralis* attacks Arabidopsis leaves, calcium levels increased and CAM1 and WRKY53 separate, allowing WRKY53 to interact with the *LOX3* promoter. In addition, we demonstrated that WRKY53 negatively regulates *LOX3* expression and the JA biosynthesis pathway (Figure 1e and Figure 3a).

## 3. Discussion

In this study, we describe how CAM1 and WRKY53 affect the resistance of Arabidopsis plants by negatively regulating the JA biosynthesis pathway (Figure 4).

We found that the *cam1* and *wrky53* mutants were more resistant to *Spodoptera littoralis* than the WT Arabidopsis plants. Due to the importance of JA in insect resistance [10], we detected the JA concentration in three plants’ leaves. Compared with the WT group, the JA concentration in the *cam1* mutant was increased, but the difference was not significant, and the JA concentration in the *wrky53* mutant was significantly increased. Higher JA concentration enables plants to make faster defense responses [31]. Thus, we explored why JA concentration increased in these two mutants.

Notably, changes in intracellular Ca^2+^ concentrations regulate the JA biosynthesis pathway. When plants are wounded by insects, CML37 and CML42 influence the JA biosynthesis pathway [3,4]. Based on the results in Figure 1, we speculate that CAM1 may also be involved in regulating the JA biosynthesis pathway. Additionally, we found that CAM1 was more closely related to WRKY53 by screening several WRKYs (Appendix A). Because WRKYs can interact with CAMs [29,30], we reasoned that WRKY53 interacts with CAM1. Y2H, LCI, and in vitro pull-down assays confirmed this (Figure 2a–c). In addition, as WRKY53 is a group III WRKY, four conserved amino acids, 57 V, 67 S, 68 L, and 69 L, may form a fixed structure to combine with CAM1 (Figure 2d,e). Although previous studies have reported the interaction between CAMs and WRKYs, their participation in signaling pathways in plants has not been studied.

Because WRKY53 can bind to the W-box, we reasoned that WRKY53, a WRKY family transcription factor, should bind to the W-boxes in the *LOX3* and *LOX4* promoters. Indeed, an EMSA performed to verify this showed that WRKY53 bound to the W-boxes in the *LOX3* and *LOX4* promoters (Appendix A). Because of the upregulation of JA concentrations in *cam1* and *wrky53* compared to the WT (Figure 1e), we suggested that CAM1 and WRKY53 participated in JA biosynthesis because WRKY53 negatively regulated *LOX3* and *LOX4* expression. Additionally, a luciferase activity assay showed that WRKY53 is a negative regulator of *LOX3* and *LOX4*. However, this negative regulatory effect was interrupted by the addition of CAM1 (Figure 3a and Appendix A). Therefore, we speculated that CAM1 and *LOXs* promotors could competitively combine with WRKY53. Referring to the results of Appendix A, the W-box1 of *LOX3* promotor, which had the strongest binding ability with WRKY53, was used as the representative for the following study. We found that CAM1 and WRKY53 bound together when calcium was not added, while when calcium concentration increased, the protein interaction between CAM1 and WRKY53 was unbound, allowing WRKY53 to bind to the *LOXs* promoter and negatively regulating their expression (Figure 3b,c). When Arabidopsis was attacked by *Spodoptera littoralis*, calcium concentration increased in leaf cells [2]. Additionally, this was the reason why JA concentration was higher in the *wrky53* mutant (Figure 1e). For the *cam1* mutant, the results of larval length, weight gain, and JA concentration were between the data of WT and *wrky53* mutant (Figure 1). We speculated that CAM1 was the upstream of WRKY53 and might be involved in a variety of signaling pathways.

Few studies have focused on the role of WRKY53 in plant defense against *Spodoptera littoralis*. Our findings thus enrich the understanding of the functions of WRKY53. However, CAM1 may take part in other pathways to affect Arabidopsis plant resistance to insects, and other calcium sensors may also interact with WRKY53 to participate in insect resistance, which requires further research.

## 4. Materials and Methods

### 4.1. Plant Materials and Culture Conditions

Wild-type (Col-0) Arabidopsis (WT), *cam1* SALK_202076C (AT5G37780) and *wrky53* SALK_034157 (AT4G23810) were used as plant materials. The *wrky53* mutant was kindly provided by Prof. Diqiu Yu (Xishuangbanna Tropical Botanical Garden, Chinese Academy of Sciences, China). After being vernalized at 4 °C for 2 days in the dark, the seeds were sown in the autoclaved soil mixture and placed in an incubator (Percival model: I-36vl). Arabidopsis plant growth conditions are 21–23 °C, 70% relative humidity, 16 h light/8 h dark and 80–110 μmol m^−2^ s^−1^ light intensity. The plants used for insect inoculation were grown for 4 weeks and the plants used for JA measurement experiments were grown for 2 weeks.

### 4.2. Spodoptera Littoralis Egg Hatching and Inoculation

*Spodoptera littoralis* eggs were used as insect materials. Their hatched conditions are 28 ± 1 °C and 50 ± 10% relative humidity. After 1–2 days, each plant was inoculated with one larva with identical lengths and weights. Seven days later, the insect’s lengths and weights were measured. Each group contained about 30 insects.

### 4.3. Yeast Two-Hybrid (Y2H) Assay

The CDS of *CAM1* were cloned into pGADT7 with BamHI and PstI and CDS encoding amino acid residues 1–217 of *WRKY53* (WRKY53 (residues 1–217)) were cloned into pGBKT7 with EcoRI and BamHI, respectively. The plasmid pairs AD plus BK, T7 plus 53 and CAM1-AD plus WRKY53 (residues 1–217)-BK were co-transformed into competent yeast strain AH109 cells. Cell growth went through three layers of selection: SD/–Leu/–Trp, SD/–Ade/–His/–Leu/–Trp and SD/–Ade/–His/–Leu/–Trp with X-α-gal. Only cell growth on SD/–Leu/–Trp and SD/–Ade/–His/–Leu/–Trp and blue colonies on SD/–Ade/–His/–Leu/–Trp with X-α-gal indicate interactions between the two proteins.

### 4.4. Firefly Luciferase Complementation Imaging Assay (LCI)

The CDS of *CAM1* was cloned into the Cluc plasmid with BamHI and SalI, and the CDS of *WRKY53* was cloned into the Nluc plasmid with BamHI and SalI. Then, Cluc plasmid, Nluc plasmid, CAM1-Cluc plasmid and WRKY53-Nluc plasmid were transformed into Agrobacterium strain GV3101, respectively. Bacteria re-suspension with infection solution (0.213 g Mes·H_2_O, 0.203 g MgCl_2_·6H_2_O, 4 mg acetosyringone, pH 5.6–5.8, 100 mL) with OD600 0.4–0.5 was 1:1 mixed according to the groups: Cluc/Nluc, CAM1-Cluc/Nluc, Cluc/WRKY53-Nluc and CAM1-Cluc/WRKY53-Nluc. Each *Nicotiana tabacum* leaf was divided into four quadrants before injection with the four groups, respectively. After 2–3 days, fluorescence from luciferase in *Nicotiana tabacum* leaves infected with Agrobacterium was imaged with a molecular imaging system (LB983, Berthold Technologies, BadWildbad, Germany).

### 4.5. In Vitro Pull-Down Assay

The CDS of *CAM1* was cloned into pGEX-4T-1 with BamHIand SalI and the CDS of *WRKY53* was cloned into pET28A with NdeI and SalI. GST, CAM1-GST and WRKY53-HIS were transformed into *E. coli* Rosetta (DE3) cells for protein expression. Protein GST was used as the bait protein. The bindings of CAM1-GST with WRKY53-HIS were detected by immunoblot analysis using anti-GST and anti-HIS antibodies. The pulldown buffer was 1% NP40, 150 mM NaCl, 50 mM Tris–HCl, 1 mM EGTA, pH7.5.

### 4.6. Electrophoretic Mobility Shift Assay (EMSA)

The W-boxes of the *LOX3* and *LOX4* promoters were used to generate 3′-biotin-labeled probes. WRKY53-MBP and MBP (as a control) were used in EMSAs with a LightShift Chemiluminescent EMSA Kit (Thermo Fisher Scientific, Waltham, MA, USA) following the manufacturer’s protocol. The probe sequences in the EMSA were shown in Appendix A.

### 4.7. Luciferase Activity Assay

The *LOX3* and *LOX4* promoters and *WRKY53* were cloned into pGreenII 0800-LUC and pGreenII 62-SK, respectively, and the resulting plasmids were transformed into Agrobacterium strain GV3101. Each leaf was divided into four quadrants, which were injected with pGreenII 0800-LUC/pGreenII 62-SK, *LOX3 (LOX4) pro* pGreenII 0800-LUC/pGreenII 62-SK, pGreenII 0800-LUC/WRKY53 pGreenII 62-SK and *LOX3 (LOX4) pro* pGreenII 0800-LUC/WRKY53 pGreenII 62-SK, respectively. Firefly luciferase (LUC) and Renilla luciferase (REN) activities were measured using a Dual-Luciferase Reporter Gene Assay Kit with Glomax (Madison, WI, USA).

### 4.8. Measuring JA Levels in Leaves

JA levels were measured in Arabidopsis leaves in three experimental groups (WT group, *cam1* group, *wrky53* group). The internal standard was H_2_JA [32,33], and analysis was performed on an Agilent 1290 system (Agilent Technologies Co. Ltd., Palo Alto, CA, USA) with AB SCIEX-6500Qtrap (Anheuser-Busch, Saint Louis, MO, USA). Each group had three biological replicates and every replicate had more than 30 plants.

### 4.9. Statistical Analysis

Changes in JA concentration and increases in larval length and weight were examined by Dunnett’s C (variance not neat) (*p* < 0.05).

## 5. Conclusions

In this study, we revealed the role of CAM1 and WRKY53 in Arabidopsis plant defense response. In healthy plants, CAM1 interacts with WRKY53 and WRKY53 does not bind to the *LOXs* promotors. However, in insect-wounded plants, WRKY53 is isolated from the CAM1-WRKY53 complex and WRKY53 negatively regulates *LOXs* gene expression. These results suggest that WRKY53 is a negative regulator of Arabidopsis plant defense against *Spodoptera littoralis*.

## Figures and Tables

**Figure 1 ijms-23-07718-f001:**
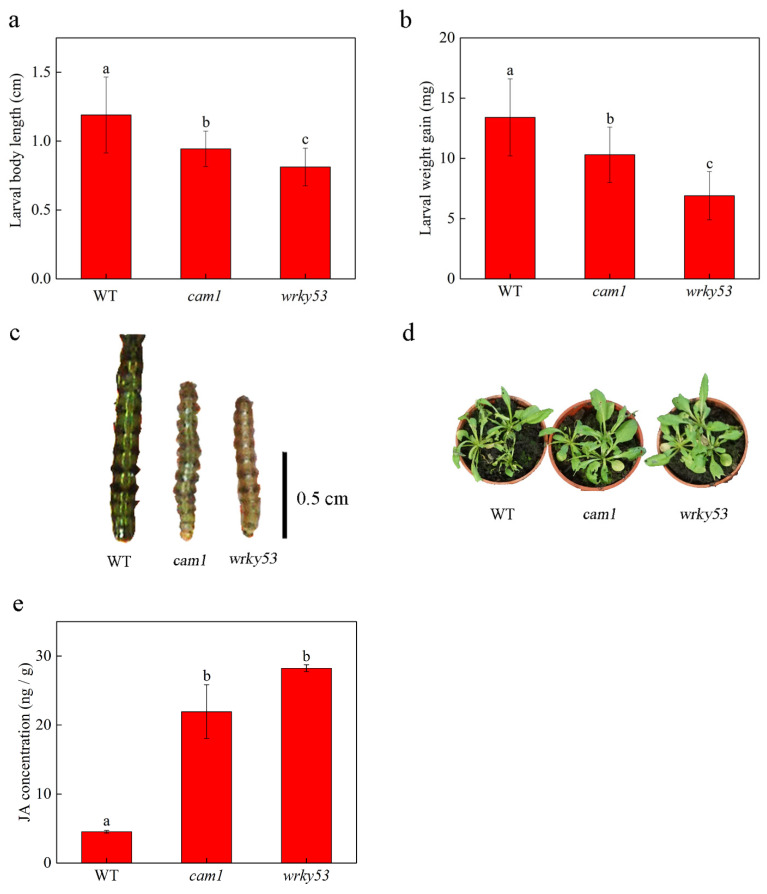
Arabidopsis plants show plant defense against *Spodoptera littoralis* in *cam1* and *wrky53* mutants. (**a**,**b**) Larval body length and weight gain were measured 7 days after inoculation. Every pot had three Arabidopsis plants, one larva was put in each plant and each group contained approximately 25–40 larvae. (**c**) Larvae phenotypes of the WT, *cam1* and *wrky53* mutants. (**d**) Conditions of plants after 7 days of larval feeding. (**e**) The JA concentration in each group, each group had three biological replicates and every replicate had more than 30 plants. In (**a**–**d**), plants were grown for 4 weeks (ten leaves), and in (**e**), plants were grown for 2 weeks. Error bars denote ± SEM, columns labeled with different letters are significantly different at *p* < 0.05, Dunnett’s C (variance not neat).

**Figure 2 ijms-23-07718-f002:**
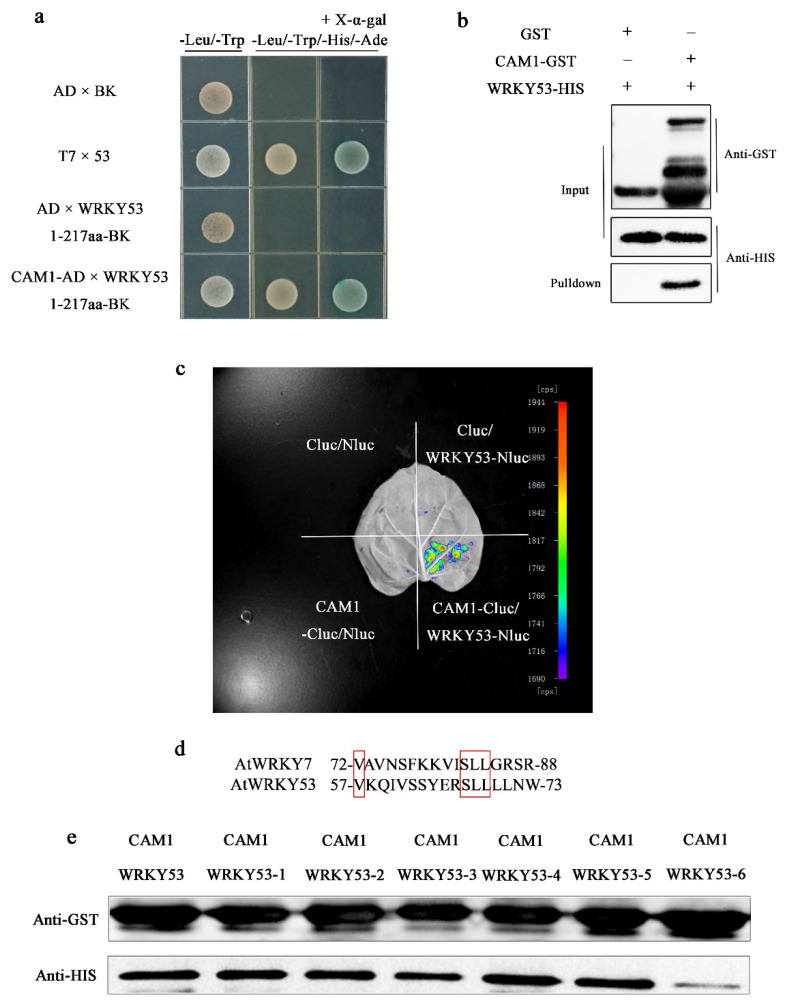
CAM1 interacts with WRKY53. (**a**–**c**) Y2H, in vitro pull-down and LCI assays, respectively, show that CAM1 interacts with WRKY53. In the Y2H assays, the N terminus of WRKY53 (residues 1–217) was used as the bait vector. (**d**) The amino acids outlined in red were conserved amino acids in WRKY53 compared to WRKY7. (**e**) WRKY53 is the unmutated protein; in WRKY53-1, 57 V was mutated to R; in WRKY53-2, 67 S was mutated to R; in WRKY53-3, 58 L was mutated to R; in WRKY53-4, 59 L was mutated to R; in WRKY53-5, 67-69 SLL was mutated to RRR; and in WRKY53-6, four amino acids were mutated to RRRR.

**Figure 3 ijms-23-07718-f003:**
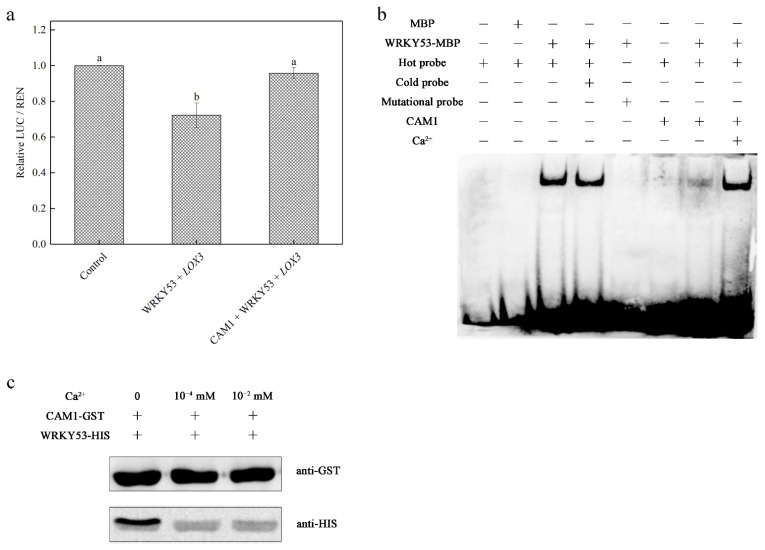
WRKY53 functions as a negative regulator and directly bind to the *LOX3* promoter. (**a**) shows that WRKY53 pGreenII 62-SK negatively affects *LOX3* expression, and CAM1 pGreenII 62-SK decreased the negative regulation. Error bars denote ± SEM, columns labeled with different letters are significantly different at *p* < 0.05, Dunnett’s C (variance not neat). (**b**,**c**) EMSA and pull-down assay showing the effects of calcium and CAM1 on the binding between WRKY53 and the first W-box in the *LOX3* promoter. In (**b**), a hot probe refers to a biotin-labeled probe and a cold probe to an unlabeled probe (200-fold the concentration of the hot probe). Ca^2+^ concentration is 10^−2^ mM.

**Figure 4 ijms-23-07718-f004:**
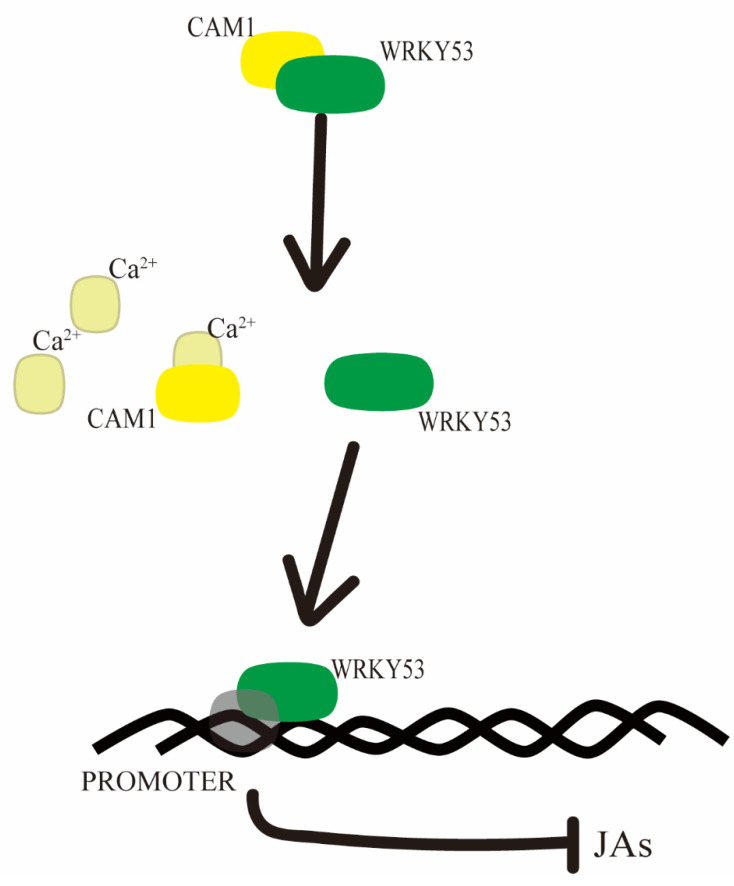
Proposed model of CAM1 and WRKY53 in JA biosynthesis in Arabidopsis. When CAM1 senses calcium, WRKY53 separates from the complex of CAM1 and WRKY53, WRKY53 binds to the promoters of *LOXs* and negatively regulates *LOXs* expression, thus reducing the JA concentration.

## Data Availability

The data that support the findings of this study are available from the corresponding author upon request.

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
