# Peer review of "CALMODULIN1 and WRKY53 Function in Plant Defense by Negatively Regulating the Jasmonic Acid Biosynthesis Pathway in Arabidopsis"

_ijms, 2022, doi:10.3390/ijms23147718_

Round 1

Reviewer 1 Report

The JA signaling pathway plays an important role in plant resistance to insect attack, but the regulation mechanism of JA signaling during plant defense remains unclear. The authors wanted to propose a new regulatory pathway of JA signaling in response to insect attack, however, some of the key data contradict with existing findings and due to flaws in their experimental design, the conclusions couldn't be supported by the data provided in the study.

Major concerns:

  1. The in vitro pull-down assay of CAM1 and WRKY53. The authors showed that CAM1 interacted with WRKY53 in vitro without calcium and the interaction is inhibited when adding calcium (Fig 3d). This contradicts with the current knowledge of the molecular mechanism of calmodulin interacting with calmodulin-binding proteins, as well as the published data on CAM-WRKY interactions. Calmodulins need to be activated by calcium before they can bind to their interactors. It is very rare that a non-active form of calmodulin but not its active form can bind to its interactor. Moreover, a few publications already showed that the CAM-WRKY interaction requires free calcium (Park et al., 2005, Popescu et al., 2007), including WRKY7, which the authors proposed that WRKY53 harbors a very similar binding domain. The pull-down assay in the paper was not properly designed to clarify this. It is unclear what buffer were used for the zero-calcium reaction but the right way to do this is to add EGTA to the pull-down buffer to make sure there is no free calcium in the reaction.
  2. The interaction of CAM1 with WRKY53 is not new as it was already confirmed in the Popescu et al., 2007 paper by protein microarrays assay and in vivo coimmunoprecipitation. WRKY53 also interacted with multiple CML proteins in the 2007 paper. The logic of choosing CAM1 but not the other CAMs or CMLs for this study is unclear. It is very likely that the interaction or co-function with WRKY53 were not specific to CAM1 and other CAMs/CMLs may also involved, especially given that cam1 had little JA increase phenotype. A better way to approach this is to screen a range of cam or cml mutants to figure out who is specific to insect resistance or whether there is functional redundancy there. In addition, to confirm that CAM1 and WRKY53 works in the same pathway, the assays in Figure 1 needs to performed in a cam1 wrky53 double mutant for verification.
  3. The JA experiment in Fig. 1e was not properly designed. It should be done in plants both with and without insect inoculation. The authors only measured steady-level JA but not insect-elicited JA. Without this data, it is hard to argue that the negative regulation of WRKY53 to JA production is calcium-dependent (through insect elicitation).
  4. A lot of the experiment details were not included or clearly described in the Methods or Results, making the interpretation of the data difficult. The reagents, plant materials and vector constructs need to be adequately described in the Methods.

Reviewer 2 Report

Jasmonic acid (JA) is an important hormone that functions in plant defense. In this study, the role of CAM1 and WRKY53 in Arabidopsis plant defense response was investigated. Results showed that WRKY53 is isolated from the CAM1-WRKY53 complex in insects wounded plants, and WRKY53 negatively regulates LOXs gene expression. The results are interesting. One suggestion is providing the regulatory model of WRKY53 function in plant defense in the paper.  

Round 2

Reviewer 1 Report

Although I am still concerned about the calcium-dependent inhibition of CAM1 binding to WRKY53, I will just accept the authors’ explanations. I appreciate that the authors did CAM1 EF region mutations and tested them in pull-down assays, however, the data is not convincing, as there seems to be opposite results by mutating different EF domains and also there is no quantitative analysis of the band intensities.

And there are still some mistakes the authors should correct in the manuscript.

1.     In the newly provided Fig. 4, the drawing in the JA production part is misleading. It should lead to inhibition of JA production rather promoting JA production. Please change the arrow to another symbol that means inhibition.  And please provide a short legend for Fig. 4 too.

2.     In the authors response letter, they said that they have EGTA in the pull-down buffer, but in the Methods part, they wrote EDTA. 
